# Learning Cross-Context Entity Representations from Text

## Abstract

Language modeling tasks, in which words, or word-pieces, are predicted on the basis of a local context, have been very effective for learning word embeddings and context dependent representations of phrases. Motivated by the observation that efforts to code world knowledge into machine readable knowledge bases or human readable encyclopedias tend to be entity-centric, we investigate the use of a fill-in-the-blank task to learn context independent representations of entities from the text contexts in which those entities were mentioned. We show that large scale training of neural models allows us to learn high quality entity representations, and we demonstrate successful results on four domains: (1) existing entity-level typing benchmarks, including a 64% error reduction over previous work on TypeNet (Murty et al., 2018); (2) a novel few-shot category reconstruction task; (3) existing entity linking benchmarks, where we match the state-of-the-art on CoNLL-Aida without linking-specific features and obtain a score of 89.8% on TAC-KBP 2010 without using any alias table, external knowledge base or in domain training data and (4) answering trivia questions, which uniquely identify entities. Our global entity representations encode fine-grained type categories, such as *Scottish footballers*, and can answer trivia questions such as *Who was the last inmate of Spandau jail in Berlin?*.

## 1 Introduction

A long term goal of artificial intelligence has been the development and population of an entity-centric representation of human knowledge. Efforts have been made to create the knowledge representation with knowledge engineers (Lenat et al., 1986) or crowdsourcers (Bollacker et al., 2008). However, these methods have relied heavily on human definitions of their ontologies, which are both limited in scope and brittle in nature. Conversely, due to recent advances in deep learning, we can now learn robust general purpose representations of words (Mikolov et al., 2013) and contextualized phrases (Peters et al., 2018) directly from large textual corpora. In particular, we observe that existing methods of building contextualized phrase representations capture a significant amount of local semantic context (Devlin et al., 2019). We hypothesize that by learning an *entity encoder* which aggregates all of the textual contexts in which an entity is seen, we should be able to extract and condense general purpose knowledge about that entity.

Consider the following *contexts* in which an entity mention has been replaced a [MASK]:

. . . the second woman in space, 19 years after [MASK].

. . . [MASK], a Russian factory worker, was the first woman in space . . .

. . . [MASK], the first woman in space, entered politics . . . .

As readers, we understand that *first woman in space* is a unique identifier, and we are able to fill in the blank unambiguously. The central hypothesis of this paper is that, by matching entities to the contexts in which they are mentioned, we should be able to build a representation for Valentina Tereshkova that encodes the fact that she was the first woman in space, that she was a politician, etc. and that we should be able to use these representations across a wide variety of downstream entity-centric tasks.

We present RELIC (Representations of Entities Learned in Context), a table of independent entity embeddings that have been trained to match fixed length vector representations of the textual context in which those entities have been seen. We apply RELIC to entity typing (mapping each entity to its properties in an external, curated, ontology); entity linking (identifying which entity is referred to by a textual context), and trivia question answering (retrieving the entity that best answers a question). Through these experiments, we show that:

- RELIC accurately captures categorical information encoded by human experts in the Freebase and Wikipedia category hierarchies. We demonstrate significant improvements over previous work on established benchmarks, including a 64% error reduction in the TypeNet low data setting. We also show that given just a few exemplar entities of a given category such as *Scottish footballers* we can use RELIC to recover the remaining entities of that category with good precision.

- Using RELIC for entity linking can match state-of-the-art approaches that make use of non-local and non-linguistic information about entities. On the CoNLL-Aida benchmark, RELIC achieves a 94.9% accuracy, matching the state-of-the-art of Raiman & Raiman (2018), despite not using any entity linking-specific features. On the TAC-KBP 2010 benchmark RELIC achieves 89.8% accuracy, just behind the top ranked system (Raiman & Raiman, 2018), which makes use of external knowledge bases, alias tables, and task-specific hand-engineered features.

- RELIC learns better representations of entity properties if it is trained to match just the contexts in which entities are mentioned, and not the surface form of the mention itself. For entity linking, the opposite is true.

- We can treat the RELIC embedding matrix as a store of knowledge, and retrieve answers to questions through nearest neighbor search. We show that this approach correctly answers 51% of the questions in the TriviaQA reading comprehension task (Joshi et al., 2017) despite not using the task's evidence text at inference time. The questions answered correctly by RELIC are surprisingly complex, such as *Who was the last inmate of Spandau jail in Berlin?*

## 2 RELATED WORK

**Entity linking** The most widely studied entity-level task is entity linking—mapping each entity mention onto a unique entity identifier. The Wikification task (Ratinov et al., 2011; Cheng & Roth, 2013), in particular, is similar to the work presented in this paper, as it requires systems to map mentions to the Wikipedia pages describing the entities mentioned. There is significant previous work that makes use of neural context and entity encoders in downstream entity linking systems (Sun et al., 2015; Yamada et al., 2016; 2017; Gupta et al., 2017; Murty et al., 2018; Kolitsas et al., 2018), but that previous work focuses solely on discriminating between entities that match a given mention according to an external alias table. Here we go further in investigating the degree to which RELIC can capture world knowledge about entities.

**Mention-level entity typing** Another well studied task is mention-level entity typing (e.g. Ling & Weld, 2012; Choi et al., 2018). In this task, entities are labeled with types that are supported by the immediate textual context. For example, given the sentence *'Michelle Obama attended her book signing'*, Michelle Obama should be assigned the type *author* but not *lawyer*. Subsequently, mention-level entity typing systems make use of contextualized representations of the entity mention, rather than the global entity representations that we focus on here.

**Entity-level typing** An alternative notion of entity typing is entity-level typing, where each entity should be associated with all of the types supported by a corpus. Yaghoobzadeh & Schütze (2015) and Murty et al. (2018) introduce entity-level typing tasks, which we describe more in Section 5.2. Entity-level typing is an important task in information extraction, since most common ontologies make use of entity type systems. Such tasks provide a strong method of evaluating learned global representations of entities.

**Using knowledge bases**   There has been a strong line of work in learning representations of entities by building knowledge base embeddings (Bordes et al., 2011; Socher et al., 2013; Yang et al., 2014; Toutanova et al., 2016; Vilnis et al., 2018), and by jointly embedding knowledge bases and information from textual mentions (Riedel et al., 2013; Toutanova et al., 2015; Hu et al., 2015). Das et al. (2017) extended this work to the SPADES fill-in-the-blank task (Bisk et al., 2016), which is a close counterpart to RELIC's training setup. However, we note that all examples in SPADES correspond to a fully connected sub-graph in Freebase Bollacker et al. (2008). Subsequently, the contents are very limited in domain and Das et al. (2017) show that it is essential to use the contents of Freebase to do well on this task. We consider the unconstrained TriviaQA task (Joshi et al., 2017), introduced in Section 5.5, to be a better evaluation for open domain knowledge representations.

**Fill-in-the-blank tasks**   There has been significant previous work in using fill-in-the-blank losses to learn context independent word representations (Mikolov et al., 2013), and context-dependent word and phrase representations (Dai & Le, 2015; Peters et al., 2018; Radford et al., 2018; Devlin et al., 2019). Cloze-style tasks, in which a system must choose which of a few entities best fill a blanked out span, have also been proposed as a method of evaluating reading comprehension (Hermann et al., 2015; Long et al., 2016; Onishi et al., 2016). For entities, Long et al. (2017) consider a similar fill-in-the-blank task as ours, which they frame as rare entity prediction. Yamada et al. (2016) and Yamada et al. (2017) train entity representations using a fill-in-the-blank style loss and a bag-of-words representation of mention contexts. Yamada et al. (2016; 2017) in particular take an approach that is very similar in motivation to RELIC, but which focuses on learning entity representations for use as features in downstream classifiers that model non-linear interactions between a small number of candidate entities. In Section 5.4, we show that Yamada et al. (2017)'s entity embeddings are good at capturing broad entity types such as *Tennis Player* but less good at capturing more complex compound types such as *Scottish Footballers*. In Section 5.1, we also show that by performing nearest neighbor search over the 818k entities in the TAC knowledge base, RELIC can surpass Yamada et al. 2017's performance on the TAC-KBP 2010 entity linking benchmark (Ji et al., 2010). This is despite the fact that Yamada et al. massively restrict the linking search space with an externally defined alias table, and incorporate task-specific hand-engineered features. On the CoNLL-Aida benchmark, we show that RELIC surpasses Yamada et al. 2017 and matches Raiman & Raiman 2018 without using any entity linking-specific features.

# 3   LEARNING FROM CONTEXT

## 3.1   RELIC TRAINING INPUT

Let $\mathcal{E} = \{e_0 \ldots e_N\}$ be a predefined set of entities, and let $\mathcal{V} = \{[\text{MASK}], [\text{E}_s], [\text{E}_e], w_1 \ldots w_M\}$ be a vocabulary of words. A *context* $\mathbf{x} = [x_0 \ldots x_l]$ is a sequence of words $x_i \in \mathcal{V}$. Each context contains exactly one entity start marker $x_k = [\text{E}_s]$ and one entity end marker $x_j = [\text{E}_e]$, where $j - k > 1$. The sequence of words between these markers, $[x_{k+1} \ldots x_{j-1}]$, is the entity mention.

Our training data is a corpus of (context, entity) pairs $\mathcal{D} = [(\mathbf{x}_0, y_0) \ldots (\mathbf{x}_N, y_N)]$. Each $y_i \in \mathcal{E}$ identifies an entity that corresponds to the single entity mention in $\mathbf{x}_i$. We train RELIC to correctly match match the entities in $\mathcal{D}$ to their mentions. We will experiment with settings where the mentions are unchanged from the original corpus, as well as settings where with some probability $m$ (the *mask rate*) all of the words in the mention have been replaced with the uninformative [MASK] symbol. We hypothesize that this parameter will play a role in the effectiveness of learned representations in downstream tasks.

For clean training data, we extract our corpus from English Wikipedia[1]. See Section 4 for details.

## 3.2   CONTEXT ENCODER

We embed each context in $\mathcal{D}$ into a fixed length vector using a Transformer text encoder (Vaswani et al., 2017), initialized with parameters from the BERT-base model released by Devlin et al. 2019. All parameters are then trained further using the objective presented below in Section 3.4.

---

[1]https://en.wikipedia.org

We take the output of the Transformer corresponding to the initial [CLS] token in BERT's sequence representation as our context encoding, and we linearly project this into $\mathbb{R}^d$ using a learned weight matrix $W \in \mathbb{R}^{d \times 768}$ to get a context embedding in the same space as our entity embeddings.

### 3.3 ENTITY EMBEDDINGS

Each entity $e \in \mathcal{E}$ has a unique and abstract Wikidata QID[2]. RELIC maps these unique IDs directly onto a dedicated vector in $\mathbb{R}^d$ via a $|\mathcal{E}| \times d$ dimensional embedding matrix. In our experiments, we have a distinct embedding for every concept that has an English Wikipedia page, resulting in 5m entity embeddings overall.

### 3.4 RELIC TRAINING LOSS

RELIC optimizes the parameters of the context encoder and entity embedding table to maximize the compatibility between observed (context, entity) pairs. Let $g(\mathbf{x}) \to \mathbb{R}^d$ be a context encoder, and let $f(e) \to \mathbb{R}^d$ be an embedding function that maps each entity to its $d$ dimensional representation via a lookup operation. We define a compatibility score between the entity $e$ and the context $\mathbf{x}$ as the scaled cosine similarity[3]

$$s(\mathbf{x}, e) = a \cdot \frac{g(\mathbf{x})^\top f(e)}{||g(\mathbf{x})|| ||f(e)||} \tag{1}$$

where the scaling factor $a$ is a learned parameter, following Wang et al. (2018a). Now, given a context $\mathbf{x}$, the conditional probability that $e$ was the entity seen with $\mathbf{x}$ is defined as

$$p(e|\mathbf{x}) = \frac{\exp(s(\mathbf{x}, e))}{\sum_{e' \in \mathcal{E}} \exp(s(\mathbf{x}, e'))} \tag{2}$$

and we train RELIC by maximizing the average log probability

$$\frac{1}{|\mathcal{D}|} \sum_{(\mathbf{x}, y) \in \mathcal{D}} \log p(y|\mathbf{x}). \tag{3}$$

In practice, the definition of probability in Equation 2 is prohibitively expensive for large $|\mathcal{E}|$ (we use $|\mathcal{E}| \approx 5M$). Therefore, we use a noise contrastive loss (Gutmann & Hyvärinen, 2012; Mnih & Kavukcuoglu, 2013). We sample $K$ negative entities from a noise distribution $p_{noise}(e)$:

$$e'_1, e'_2, \ldots, e'_K \sim p_{noise}(e) \tag{4}$$

Denoting $e'_0 := e$, we then compute our per-example loss using cross entropy:

$$l(s, \mathbf{x}, e) = -\log \frac{\exp(s(\mathbf{x}, e))}{\sum_{j=0}^{K} \exp(s(\mathbf{x}, e'_j))} \tag{5}$$

In practice, we train our model with minibatch gradient descent and use all other entries in the batch as negatives. That is, in a batch of size 4, entities for rows 1, 2, 3 will be used as negatives for row 0. This is roughly equivalent to $p_{noise}(e)$ being proportional to entity frequency.

## 4 EXPERIMENTAL SETUP

To train RELIC, we obtain data from the 2018-10-22 dump of English Wikipedia. We take $\mathcal{E}$ to be the set of all entities in Wikipedia (of which there are over 5 million). For each occurrence of a hyperlink, we take the context as the surrounding sentence, replace all tokens in the anchor text with a single [MASK] symbol with probability $m$ (see Section 5.3 for a discussion of different masking rates) and set the ground truth to be the linked entity. We limit each context sentence to 128 tokens. In this way, we collect a high-quality corpus of over 112M (context, entity) pairs. Note in particular that an entity never co-occurs with text on its own Wikipedia page, since a page will not hyperlink to itself. We set the entity embedding size to $d = 300$.

We train the model using TensorFlow (Abadi et al., 2016) with a batch size of 8,192 for 1M steps on Google Cloud TPUs.

---

[2]https://www.wikidata.org/wiki/Q43649390

[3]In our experiments, we found cosine similarity to be more effective than dot product.

| System | CoNLL-Aida | TAC-KBP 2010 |
|---|---|---|
| Sil et al. 2018 | 94.0 | 87.4 |
| Yamada et al. 2016 | 91.5 | 85.5 |
|    - entity linking features | 81.1 | 80.1 |
| Yamada et al. 2017 | 94.3 | 87.7 |
| Radhakrishnan et al. 2018 | 93.0 | 89.6 |
| Raiman & Raiman 2018 | 94.9 | 90.9 |
| RELIC | 81.9 | 87.5 |
| RELIC + CoNLL-Aida tuning | 94.9[4] | 89.8 |

Table 1: RELIC achieves comparable results to best performing dedicated entity-linking systems despite using no external resources or task specific features. When given a standard CoNLL-Aida alias table and tuned on the CoNLL-Aida training set, RELIC's learned representations match the state-of-the-art DeepType system which relies on the large hand engineered Wikidata knowledge base.

## 5   EVALUATION

We evaluate RELIC's ability to: (1) solve the entity linking task without access to any task specific alias tables or features; (2) accurately capture entity properties that have been hand-coded into TypeNet and Wikipedia categories; (3) capture trivia knowledge specific to individual entities.

First we present results on established entity linking and entity typing tasks, to compare RELIC's performance to established baselines and we show that the choice of masking strategy (Section 3) has a significant and opposite impact on performance on these tasks. We hypothesize that RELIC is approaching an upper bound on established entity-level typing tasks, and we introduce a much harder category completion task that uses RELIC to populate complex Wikipedia categories. We also apply RELIC's context encoder and entity embeddings to the task of end-to-end trivia question answering, and we show that this approach can capture more than half of the answers identified by the best existing reading comprehension systems.

### 5.1   ENTITY LINKING

RELIC can be used to directly solve the entity linking problem. We just need to find the single entity that maximizes the cosine similarity in Equation 1 for a given context. For the entity linking task, we create a context from the document's first 64 tokens as well as the 64 tokens around the mention to be linked. This choice of context is well suited to the documents in the CoNLL-Aida and TAC-KBP 2010 datasets, since those documents tend to be news articles in which the introduction is particularly information dense. In Table 1 we show performance for RELIC in two settings. First, we report the accuracy for the pure RELIC model with no in-domain tuning. Then, we report the accuracy for a RELIC model that has been tuned on the CoNLL-Aida training set. On the CoNLL-Aida benchmark, we also adopt a standard alias table (Pershina et al., 2015) for this tuned model, as is commonly done in previous entity linking work.

It is clear that for the CoNLL-Aida benchmark in-domain tuning is essential. We hypothesize that this is because of the dataset's bias towards certain types of news content that is very unlike our Wikipedia pre-training data—specifically sports reports. However, when we do adopt the standard CoNLL-Aida training set and alias table, RELIC matches the state of the art on this benchmark, despite using far fewer hand engineered resources (Raiman & Raiman (2018) use the large Wikidata knowledge base to create entity representations). We do not make use of the TAC-KBP 2010 training set or alias table, and we observe that RELIC is already competitive without these enhancements[5]

It is significant that RELIC matches the performance of Raiman & Raiman (2018), which uses the large hand engineered Wikidata knowledge base to represent entities. This supports our central hypothesis that it is possible to capture the knowledge that has previously been manually encoded in knowledge bases, using entity embeddings learned from textual contexts alone. In the rest of

---

[4]Our finetuned CoNLL result uses the alias table of Pershina et al. (2015) at inference time.

[5]We do reduce the candidate set from the 5m entities covered by RELIC to the 818k entities in the TAC-KBP 2010 knowledge base to avoid ontological misalignment.

| System | F1 | P@1 | Acc |
|---|---|---|---|
| Yaghoobzadeh et al. 2018 | 82.3 | 91.0 | 56.5 |
| RELIC | 87.9 | 94.8 | 68.3 |
| RELIC with 5% of FIGMENT training data | 83.3 | 90.9 | 59.3 |

Table 2: Performance on FIGMENT. We report P@1 (proportion of entities whose top ranked types are correct), Micro F1 aggregated over all (entity, type) compatibility decisions, and overall accuracy of entity labeling decisions. RELIC outperforms prior work, even with only 5% of the training data.

| System | TypeNet | TypeNet - Low Data (5%) |
|---|---|---|
| Murty et al. 2018 | 78.6 | 58.8 |
| RELIC | 90.1 | 85.3 |

Table 3: Mean Average Precision on TypeNet tasks. RELIC's gains are particularly striking in the low data setting from Murty et al. (2018).

this section, we will show further support for our hypothesis by recreating parts of the Freebase and Wikipedia ontologies, and by using RELIC to answer trivia questions.

Finally, we believe that RELIC's entity linking performance could be boosted even higher through the adoption of commonly used entity linking features. As shown in Table 1, Yamada et al. (2016) use a small set of well chosen discrete features to increase the performance of their embedding based approach by 10 points. These features could be simply integrated into RELIC's model, but we consider them to be orthogonal to this paper's investigation of purely learned representations.

## 5.2 ENTITY-LEVEL FINE TYPING

We evaluate RELIC's ability to capture entity properties on the FIGMENT (Yaghoobzadeh & Schütze, 2015) and TypeNet (Murty et al., 2018) entity-level fine typing tasks which contain 102 and 1,077 types drawn from the Freebase ontology (Bollacker et al., 2008). The task in both datasets is to predict the set of fine-grained types that apply to a given entity. We train a simple 2-layer feed-forward network that takes as input RELIC's embedding $f(e)$ of the entity $e$ and outputs a binary vector indicating which types apply to that entity.

Tables 2, 3 show that RELIC significantly outperforms prior results on both datasets. For FIGMENT, Yaghoobzadeh et al. (2018) is an ensemble of several standard representation learning techniques: `word2vec` skip-gram contexts (Mikolov et al., 2013), *structured* skip-gram contexts (Ling et al., 2015), and `FastText` representations of the entity names (Bojanowski et al., 2017). For TypeNet, Murty et al. (2018) aggregate mention-level types and train with a structured loss based on the TypeNet hierarchy, but is still outperformed by our flat classifier of binary labels. We expect that including a hierarchical loss is orthogonal to our approach and could improve our results further.

The most striking results in Tables 2 and 3 are in the low data settings. On the low-data TypeNet setting of Murty et al. (2018), RELIC achieves a 63% error reduction over previous work, while RELIC also matches Yaghoobzadeh et al. 2018's results on FIGMENT with 5% of the training data.

## 5.3 EFFECT OF MASKING

In Section 3 we introduced the concept of masking entity mentions, and predicting on the basis of the context in which they are discussed, not the manner in which they are named. Figures 1 and 2 show the effect of training RELIC with different mask rates. It is clear that masking mentions during training is beneficial for entity typing tasks, but detrimental for entity linking. This is in accordance with our intuitions. Modeling mention surface forms is essential for linking, since these mentions are given at test time and names are extremely discriminative. However, once the mention is known the model only needs to distinguish between different entities with the same name (e.g. *President Washington, University of Washington, Washington State*) and this distinction rarely requires deep knowledge of each entity's properties. Subsequently, our best typing models are those that are forced to capture more of the context in which each entity is mentioned, because they are not allowed to

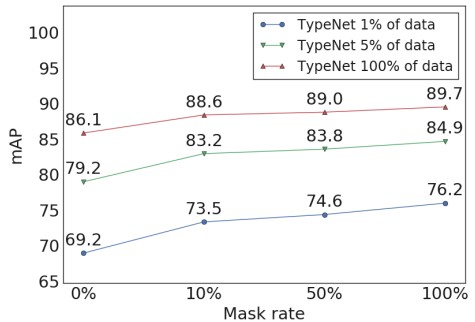

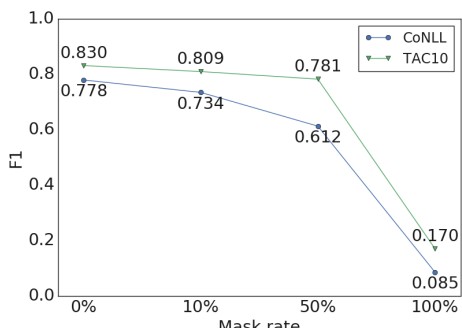

Figure 1: TypeNet entity-level typing mAP on the development set for RELIC models trained with different masking rates. A higher mask rate leads to better performance, both in low and high-data situations.

Figure 2: Entity linking accuracy for RELIC models trained with different masking rates. No alias table nor in-domain fine-tuning is used. Higher mask rates lead to worse downstream performance in entity-linking tasks.

|  | Yamada Subset | | All Entities | |
| --- | --- | --- | --- | --- |
|  | TypeNet | Wikipedia | TypeNet | Wikipedia |
| # Entities | 291,663 | 707,588 | 323,347 | 3,667,933 |
| Random | 2.7 | 0.1 | 2.5 | 0.1 |
| Yamada et al. 2017 | 25.9 | 8.0 | – | – |
| RELIC | 27.8 | 21.0 | 29.3 | 13.8 |

Table 4: Mean average precision on exemplar-based category completion (Section 5.4). The Yamada subset is filtered to only contain entities that are covered by Yamada et al. 2017, and categories are filtered to those which contain at least 300 entities (131 categories). For the "All Entities" setting, we use all Wikipedia entities covered by RELIC, and filter to categories which contain at least 1000 entities (1083 categories). The embeddings learned by Yamada et al. 2017 are competitive with RELIC on the task of populating TypeNet categories, but they are much worse at capturing the complex, and compound, typing information present in Wikipedia categories.

rely on the mention itself. The divergence between the trends in Figures 1 and 2 suggests that there may not be one set of entity embeddings that are optimum for all tasks. However, we would like to point out that that a mask rate of 10%, RELIC nears optimum performance on most tasks. The optimum mask rate is an open research question, that will likely depend on entity frequency as well as other data statistics.

## 5.4 FEW-SHOT CATEGORY COMPLETION

The entity-level typing tasks discussed above involve an in-domain training step. Furthermore, due to the incompleteness of the the FIGMENT and TypeNet type systems, we also believe that RELIC's performance is approaching the upper bound on both of these supervised tasks. Therefore, to properly measure RELIC's ability to capture complex types from fill-in-the-blank training alone, we propose:

1. a new category completion task that does not involve any task specific optimization,
2. a new Wikipedia category based evaluation set that contains much more complex compound types, such as *Scottish footballers*,

We use this new task to compare RELIC to the embeddings learned by Yamada et al. 2017.

In the new category completion task, we represent each category by randomly sampling three exemplar entities, and calculating the centroid of their RELIC embeddings. We then rank all other entities according to their dot-product with this centroid, and report the mean average precision (MAP) of the resultant ranking.

|                                       | Open-domain Unfiltered | Verified Web |
|---------------------------------------|------------------------|--------------|
| Classifier baseline (Joshi et al., 2017) | —                   | 30.2         |
| SLQA (Wang et al., 2018b)             | —                      | 82.4         |
| RELIC                                 | 35.7                   | 51.2         |
| ORQA (Lee et al., 2019)               | 45.0                   | —            |

Table 5: Answer exact match on TriviaQA. RELIC's fast nearest neighbor search over entities achieves 80% of the performance of ORQA, which runs a BERT-based reading comprehesion model over multiple retrieved evidence passages. Unlike ORQA and RELIC, the classifier baseline and SLQA have access to a single evidence document that is known to contain the answer. As a result they are solving a much easier task.

First, we apply this evaluation to the TypeNet type system introduced in (Murty et al., 2018). These types are well-curated, but tend to represent high-level categories. To measure the degree to which our entity embeddings capture finer grained type information, we construct an aditional dataset based on Wikipedia categories[6]. These tend to be compound types, such as *Actresses from London*, which capture many aspects of an entity—in this case gender, profession, and place of birth.

From Table 4 we can see that the embeddings introduced by Yamada et al. 2017 approach RELIC's performance on the TypeNet completion task, but they significantly underperform RELIC in completing the more complex Wikipedia categories. Figure 3a shows example reconstructions for randomly sampled Wikipedia categories, two from TypeNet and three from Wikipedia. Both models achieve high precision on TypeNet categories, but on the finer-grained Wikipeida categories, the Yamada et al. (2017) model tends to produce more broadly-related entites, whereas the RELIC embeddings capture entities which are much closer to the exemplars. In fact, we identify several false negatives in these examples.

## 5.5 TRIVIA QUESTION ANSWERING

Our final experiment tests RELIC's ability to answer trivia questions – which can be considered high precision categories that only apply to a single entity – using retrieval of encoded entities. TriviaQA (Joshi et al., 2017) is a question-answering dataset containing questions sourced from trivia websites, and the answers are usually entities with Wikipedia pages. The standard TriviaQA setup is a reading comprehension task, where answers are extracted from evidence documents. Here, we answer questions in TriviaQA *without access to any evidence documents at test time.*

**Model and training**   Given a question, we apply the context encoder $g$ from Section 3.4, and retrieve 1 out of 5M entities using cosine similarity. For training, we initialize both $g$ and $f$ from RELIC training. We tune only $g$'s parameters by optimizing the loss in Equation 5 applied to (question, answer entity) pairs, rather than the (context, entity) pairs seen during RELIC's training.

**Results**   TriviaQA results are shown in Table 5, and randomly sampled RELIC predictions are illustrated in Figure 3b. All systems other than RELIC in Table 5 have access to evidence text at inference time. In the open domain unfiltered setting, ORQA (Lee et al., 2019) retrieves this text from a cache of Wikipedia. In the more standard verified-web reading comprehension task, the classifier baseline and SLQA are provided with a single document that is known to contain the answer.

We consider ORQA to be the most relevant point of comparison for RELIC. We observe that the retrieve-then-read approach taken by ORQA outperforms the direct answer retrieval approach taken by RELIC. However, ORQA runs a BERT based reading comprehension model over multiple evidence passages at inference time and we are encouraged to see that RELIC's much faster nearest neighbor lookup captures 80% of ORQA's performance.

It is also significant that RELIC outperforms Joshi et al. (2017)'s reading comprehension baseline by 20 points, despite the fact that the baseline has access to a single document that is known to contain the answer. However, RELIC is still far behind the reading comprehension upper bound set

---

[6]We use the Yago 3.1 (Mahdisoltani et al., 2013) dump of extracted categories that cover at least 1,000 entities, resulting in 1,083 categories.

| Category and Exemplars | Yamada et al. 2017 | RELIC |
|---|---|---|
| tennis_player
David Goffin
Yves Allegro
Flavia Pennatta | 1. **Ekaterina Makarova**
2. **Vera Zvonareva**
3. **Flavia Pennetta**
4. **Max Mirnyi**
5. **Lisa Raymond**
AP=71.87 | 1. **Prakash Amritraj**
2. **Marco Chiudinelli**
3. **Marc Gicquel**
4. **Marius Copil**
5. **Benjamin Balleret**
AP=56.87 |
| exhibition_producer
Toledo Museum of Art
Egyptian Museum
San Jose Museum of Art | 1. **Smithsonian American Art Museum**
2. **Honolulu Museum of Art**
3. **Brooklyn Museum**
4. **Whitney Museum of American Art**
5. **Hirshhorn Museum and Sculpture Garden**
AP=38.41 | 1. **Cleveland Museum of Art**
2. **Smithsonian American Art Museum**
3. **Indianapolis Museum of Art**
4. **Cincinnati Art Museum**
5. **Museum of Fine Arts, Boston**
AP=52.52 |
| Scottish footballers
Pat Crerand
Gerry Britton
Jim McLean | 1. Ayr United F.C.
2. Clyde F.C.
3. Scottish League Cup
4. Stranraer F.C.
5. Arbroath F.C.
AP=4.57 | 1. **Tommy Callaghan**
2. **Gordon Wallace**
3. *David White***
4. Davie Dodds
5. **John Coughlin**
AP=67.10 |
| Number One Singles in Germany
Lady Marmalade
Just Give Me a Reason
I'd Do Anything for Love (But I Won't Do That) | 1. Billboard Hot 100
2. Grammy Award for Best Female Pop Vocal Performance
3. Dance Club Songs
4. Pop 100
5. Hot Latin Songs
AP=4.14 | 1. Try (Pink song)
2. *Whataya Want from Me***
3. Fuckin' Perfect
4. Beautiful (Christina Aguilra song)
5. Raise Your Glass
AP=4.59 |
| 2010 Albums
This Is the Warning
Tin Can Trust
Bionic (Christina Aguilera album) | 1. FAA airport categories
2. Rugby league county cups
3. Digital Songs
4. Country Airplay
5. Swiss federal election, 2007
AP=0.04 | 1. *All I Want Is You***
2. Don't Mess with the Dragon
3. *Believe (Orianthi album)***
4. Sci-Fi Crimes
5. **Interpol**
AP=6.78 |

(a) Top 5 predictions for a set of randomly selected categories, given 3 exemplars. The first two categories come from TypeNet, and the second two from our Wikipedia categorization dataset. Correct predictions are bolded. Predictions which are judged by the authors to be false negatives (predictions which properly belong to the target category) are indicated with asterisks**.

---

**Q:** Who was the last inmate of Spandau jail in Berlin?
A: **1. Rudolf Hess** 2. Adolf Hitler 3. Hermann Gring 4. Heinrich Himmler 5. Ernst Rhm

**Q:** Which fashionable London thoroughfare, about three quarters of a mile (1.2 km) long, runs from Hyde Park Corner to Marble Arch, along the length of the eastern side of Hyde Park?
A: **1. Park Lane** 2. Piccadilly 3. Knightsbridge 4. Leicester Square 5. Tottenham Court Road

**Q:** In which Lake District town would you find the Cumberland Pencil Museum?
A: **1. Keswick** 2. Hawkshead 3. Grasmere 4. Cockermouth 5. Ambleside

**Q:** The Wimbledon tennis tournament is held at which tennis club in London?
A: 1. Queen's Club **2. All England Lawn Tennis and Croquet Club** 3. Wimbledon Championships 4. Stade Roland-Garros 5. Wentworth Club

---

(b) TriviaQA predictions from retrieval. Questions are randomly sampled, and top 5 ranking answers are shown. Correct answer in bold. Note that even when the model is wrong, the predictions are all of the correct type.

Figure 3: Random example predictions drawn from category completion, and TriviaQA tasks.

by Wang et al. (2018b) and there is a long way to go before RELIC's embeddings can capture all of the facts that can be identified by question-dependent inference time reading.

## 6 CONCLUSION

In this paper, we demonstrated that the RELIC fill-in-the-blank task allows us to learn context independent representations of entities with their own latent ontology. We show successful entity-level typing results on FIGMENT (Yaghoobzadeh & Schütze, 2015) and TypeNet (Murty et al., 2018),

even when only training on a small fraction of the task-specific training data. We then introduce a novel few-shot category reconstruction task and when comparing to Yamada et al. (2017), we found that RELIC is better able to capture complex compound types. Our method also proves successful for entity linking, where we match the state of the art on CoNLL-Aida despite not using linking-specific features and fare similarly to the best system on TAC-KBP 2010 despite not using an alias table, any external knowledge bases, linking-specific features or even in-domain training data. Finally, we show that our RELIC embeddings can be used to answer trivia questions directly, without access to any evidence documents. We encourage researchers to further explore the properties of our entity representations and BERT context encoder, which we will release publicly.

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
