# OpenReview forum: "Learning Cross-Context Entity Representations from Text"
_ICLR.cc/2020/Conference — Reject_

### Official Review · AnonReviewer1 · 2019-10-23
**Official Blind Review #1**

**Rating:** 3

**Review:**

The paper describes a new way of learning context dependent entity representations that are capable of encoding fine-grained entity types. This is realised by matching entities to all their contexts and thereby encoding all their properties.
The contribution is RELIC, a table of entity representations that are learned given the above objective. This table, RELIC, can be used in various tasks like entity typing, entity linking and question answering. Given a corpus of entities and their respective contexts, the probability of an entity occurring under its context is maximised (additionally, the probability is minimised for negative samples). The data is taken from a Wikipedia dump. Without adding further information, the RELIC table can directly be used for entity linking.
In the entity linking task, RELIC achieves a lower precision at two benchmarks (CoNLL-Aida, TAC-KBP 2010) than other approaches. This can be tackled by fine-tuning the RELIC table, which is done with the training set from the CoNLL-Aida task. With fine-tuning, comparable results are achieved.
For entity typing, RELIC embeddings of entities are used as input for a 2-layer FF network, which then outputs which types belong to the entity. The FIGMENT and TypeNet datasets are used as benchmark here and it is shown that the RELIC based approach outperforms the other approaches.
RELIC is also used for a category completion task, where a category is represented by a centroid of three randomly sampled entities belonging to it. The entities from RELIC are then ranked with the dot-product with the centroid. Two tasks are used to measure the performance here: the TypeNet completion task and a Wikipedia-based task. Compared to embeddings of a 2017 approach, the RELIC embeddings perform better.
Finally, a QA task is performed using RELIC. Here, the questions are modelled as contexts and the entity which is closest in terms of cosine similarity is taken as answer. The performance is below the upper bound from 2018, but better than a classifier system from 2017.

Overall, the paper is well structured and manages to show that RELIC is a versatile tool on which various tasks can be performed. The technical details are well covered and the evaluation is done in a detailed way.
The contribution of the paper is mostly in the definition of context-entity pairs as input to a transformer model and in empirical evaluations. I specifically found the findings in 5.3 interesting. The technological contribution is rather limited.

Minor comments: At page 3, chapter 3.1 there is a word duplication […]to correctly match match[…] that should be corrected. Additionally, on page 5 it should be mentioned that the values in table 1 are precision values.

**Experience Assessment:**

I have published in this field for several years.

**Review Assessment: Checking Correctness Of Derivations And Theory:**

I assessed the sensibility of the derivations and theory.

**Review Assessment: Checking Correctness Of Experiments:**

I assessed the sensibility of the experiments.

**Review Assessment: Thoroughness In Paper Reading:**

I read the paper at least twice and used my best judgement in assessing the paper.

---

> ### Author Response · Authors · 2019-11-15
> **Response to reviewer #1**
>
> Thank you for your thorough reading of the paper and detailed review.
>
> We agree that this paper's modelling contributions are simple, but we also feel that our empirical findings are the main contribution of this paper. These go much further than all previous efforts in testing the limits of entity embeddings learned from textual context.
>
> As a point of clarification, we would like to point out that for the TriviaQA question answering task, RELIC is solving a much harder problem than the 2018 upper bound. RELIC has to retrieve the single correct answer from all 5m entities that have a Wikipedia page, whereas the 2018 upper bound is performing reading comprehension on a single document that is known to contain the answer (web setting).
>
> We have updated the paper with a more recent and relevant comparison to "Latent Retrieval for Weakly Supervised Open Domain Question Answering" - Kenton Lee et.al. 2019 (https://www.aclweb.org/anthology/P19-1612/). Lee et.al.'s approach does still apply a reading comprehension model, but passages that are retrieved from Wikipedia rather than the document provided in TriviaQA's reading comprehension task.
>
> RELIC's answer retrieval approach achieves 80% of the performance of Lee et.al's approach. We consider this significant, since Lee et.al. rely on an expensive BERT based co-encoding of the question and evidence text at inference time, while RELIC performs a simple nearest neighbour search over 5m pre-indexed answer candidates, which is vastly faster.
>
> We believe that our results: that RELIC can reconstruct complex Wikipedia categories; and that RELIC can directly retrieve answers to trivia questions with 80% of the accuracy of Lee et.al's retrieve-then-read model, are significant and worthy of publication at ICLR despite the simplicity of the RELIC model.
>
> We would also like to alert you to our modification of our entity linking experiments in response to questions from other reviewers. By simply reducing RELIC's search space to be more in line with other entity linking work, and adding extra document context to RELIC's text encoder, we have now matched the DeepType entity linking state of the art. DeepType relies on the large hand created Wikidata knowledge base for entity representations. The fact that RELIC can match DeepType's performance supports our central hypothesis that RELIC is managing to capture the type of knowledge that has previously been encoded by hand in structured knowledge bases.  We discuss these modifications fully in the response to all reviewers.

---

### Official Review · AnonReviewer2 · 2019-10-24
**Official Blind Review #2**

**Rating:** 6

**Review:**


This paper aims to learn entity representations by aggregating all the contexts that an entity appears in based on English Wikipedia.

The idea is very simple, basically it represents each entity as a vector, and also represents each context as a vector, and maximizes the cosine similarity between the two vectors using a negative-sampling training objective. The training process is similar as word2vec, and they leveraged all the hyperlinks in Wikipedia, and used BERT to encode the context (instead of learning a context vector for each word).

As a result, the paper demonstrates that this set of entity embeddings are highly useful, and they were evaluated on 1) entity-level typing 2) entity linking 3) few-shot category reconstruction 4) answering trivia questions (TriviaQA).

I think this is a nice empirical paper and the experiments are thorough. If they are going to release the entity embeddings, that would also benefit the community a lot and also encourage more research in this direction.

I am a bit concerned about the novelty of the approach. It is a bit surprising that nobody has experimented with this before. It seems that Yamada et al took a very similar approach but used simple bag-of-words approaches to encode the context (instead of BERT).  To me, this paper may be better for the NLP community but it should be fine to the ICLR community too.

I am also not completely sure how strong the evaluation results are indeed, esp. related to the comparisons with Yamada et al, 2017, given the approaches are similar.

- It seems to be on par with or slightly worse than Yamada et al 2017 on entity linking.

- For the category completion task, RELIC is doing much better than Yamada et al 2017 on more complex Wikipedia categories but I am not sure if it is a completely fair comparison. It’d be great if the authors can discuss all the key differences between the two approaches, from the model design to all the experimental details, that would help clear out all these confusions. I have read the related work section but can't figure out all the details.

- The entity typing results are very strong. I also like the TriviaQA experiments but the numbers are way behind the standard reading comprehension results.


Minor comment:
- Why sec 5.3 (effect of masking) is listed together with other evaluation tasks? Isn’t better to move it to analyses/ablation studies?


**Experience Assessment:**

I have published one or two papers in this area.

**Review Assessment: Checking Correctness Of Derivations And Theory:**

I carefully checked the derivations and theory.

**Review Assessment: Checking Correctness Of Experiments:**

I assessed the sensibility of the experiments.

**Review Assessment: Thoroughness In Paper Reading:**

I read the paper thoroughly.

---

> ### Author Response · Authors · 2019-11-15
> **Response to reviewer #2**
>
> Thank you for your response and careful questions. We agree that the RELIC model is simple and that the value of this paper is in the experiments which demonstrates the utility of these entity embeddings for a variety of tasks beyond entity linking. We do plan to release the table of RELIC embeddings and we hope that these will be useful to the community in much the same way that word embeddings have been.
>
> It is correct that RELIC's performance on TriviaQA is far behind that of the standard reading comprehension systems that have access to documents which are known to contain the answer. However, RELIC is solving a much harder task and to provide a more meaningful benchmark, we have updated the paper with a comparison to Lee et.al. 2019's results on the open domain version of TriviaQA. We discuss this comparison fully in the response to all reviewers and we are happy to say that RELIC's fast inner product search over 5m candidate answers captures 80% of the performance of Lee et.al.'s method, which relies on inference time reading of evidence text using an expensive BERT based model.
>
> To perform the comparison with Yamada et.al. we downloaded embeddings from their website (https://github.com/studio-ousia/ntee), filtered the entities in the Wikipedia category completion task to only those covered by Yamada et.al.'s embedding table, and then ran the same evaluation script that was used to evaluate the RELIC embeddings. As there is no task specific training for the Wikipedia category completion task, we believe that this is the cleanest comparison that we could make between the RELIC embedding table and the downloaded embeddings.
>
> You make the valid point that the masking ablations are somewhat out of place in the primary results section. However, we would like to point out that, despite the obvious simplicity of this experiment, we do believe that it is both novel and crucial to RELIC's success at entity typing and question answering. All previous related work, such as that of Yamada et.al., has been geared toward entity linking and it subsequently does not experiment with mention masking (ignoring entity names is a ridiculous decision in the context of entity linking). Conversely, for entity typing and question answering---where the name is by definition absent---entity masking is essential. Simple as it is, we consider this experiment to be a key contribution of the paper.
>
> Finally, we would like to alert you to the modifications that we have made to the entity linking experiments, to provide a cleaner comparison to previous work. By simply reducing the search space to be more in line with other work, and also expanding the context available to the RELIC model, we now match the state of the art system which relies on a large hand engineered knowledge base. Our modifications, and their significance are discussed fully in the response to all reviewers, as well as the updated paper.
>
> —————————
>
> Latent Retrieval for Weakly Supervised Open Domain Question Answering
> Kenton Lee, Ming-Wei Chang, Kristina Toutanova (2019)
> https://www.aclweb.org/anthology/P19-1612/

---

### Official Review · AnonReviewer3 · 2019-10-25
**Official Blind Review #3**

**Rating:** 3

**Review:**

This paper proposes to learn entity representations by matching entities to the context it occurs in. It also shows that using these representations is very effective for a wide variety of down-stream entity-centric tasks such as entity typing, linking, and answering entity centric trivia questions. They train the model using a corpus of entity linked Wikipedia contexts (sentences unto length 128 tokens). The context is encoded with a BERT model and the CLS representation is used as the representation of context. After obtaining the representation, they train the entity embedding (present in the sentence) to be similar to the context embedding. They test their embeddings on few down-stream entity-centric tasks — linking, typing and trivia question answering.

Strengths:
1. They try the entity representations on a wide variety of entity-centric tasks and get reasonable results.

Weaknesses:
1. The biggest weakness of the paper is wrt novelty. Masking out entities and training to context is not a new idea. As pointed by the paper, Yamada et al., 2017 have a very similar objective and it is not very clear from the paper what is the additional contribution that this paper makes. Is using pretrained LMs the major difference? If not, it would have been nice to see Yamada et al’s results with BERT. Over all, this paper needs to make its own contribution clear compared to Yamada et al., 2017.
2. The paper needs to be written more clearly at several places. Few examples are, even though in entity linking results (Table 1) the model achieves 83.0 with other papers achieving 90.9. I didnt see a discussion on how to close the gap. Even in the coNLL benchmark, the initial results of the paper is significantly behind. Claims like “CoNLL -Aida is known to be restricted and idiotic-synctatic domain” should be backed by detailed analysis or atleast a citation. Even after finetuning on the CoNLL benchmark, the result is 2.2 points behind state of the art and no discussions have been provided. As a result, I think the entity linking section needs major re-writing and explanation of the results.
3. The paper makes an interesting observation that masking of entities is better for typing tasks and it affects linking performance, because spelling features are really important for linking. It would be interesting to see a discussion on what could be done to remedy this. Because if we have to retrain entity embeddings for different tasks, then it goes against the hypothesis of the paper which is to use entity representations for a wide variety of down-stream tasks.
4. I found it confusing to read the setup in sec 5.4. especially where it says we represent each category with three random exemplars. Initially I thought 3 randomly sampled entities formed a category, which didnt make sense, but from figure 3, I think I understood that you first pick a category from Typenet and Wikipedia and then 3 entities are sampled from there. Is that correct? Regarding the results, can the poor results of Yamada et al., can be understood by the fact that it was trained using smaller number of categories? Also, why are the numbers wrt All entities left blank in Table 4. Given that your model is similar, I am assuming its easy to retrain Yamada et al and test it on the all entities benchmark?

**Experience Assessment:**

I have published one or two papers in this area.

**Review Assessment: Checking Correctness Of Derivations And Theory:**

N/A

**Review Assessment: Checking Correctness Of Experiments:**

I carefully checked the experiments.

**Review Assessment: Thoroughness In Paper Reading:**

N/A

---

> ### Author Response · Authors · 2019-11-15
> **Response to reviewer #3**
>
> Thank you for your review and detailed questions.
>
> As discussed in the response to all reviewers, we agree that RELIC's model architecture is not particularly novel. We believe that this paper's contribution is in the extensive, and novel, experiments that go well beyond previous work in testing the extent to which embeddings learned from textual context can capture the knowledge required for a wide range of entity-centric tasks.
>
> In response to your questions about our entity linking results we have updated our experiments to be more in line with other entity linking approaches. First, we have adopted the same CoNLL alias table used by most other recent approaches. Second, we reduced the TAC-KBP entity candidate set to only those entities in the TAC-KBP knowledge base (a reduction of 5m -> 818k candidates). Third we have updated provided the RELIC model with the start of each document, as well as the immediate context surrounding each entity mention.
>
> Together, these modifications have brought RELIC's performance up to match the state of the art on CoNLL (94.9%), and second only to DeepType on TAC-KBP (RELIC: 89.8%, DeepType: 90.9%). DeepType relies on the large Wikidata knowledge base for entity representations and we consider it significant that RELIC can match this system's performance with embeddings learned purely from context. We discuss our modifications more in the response to all reviewers, as well as the updated paper.
>
> As well as reporting the performance of the RELIC model that has been tuned toward the entity linking task (CoNLL + Aida tuning), we do still report the performance of the pure RELIC model that has never seen any in-domain data, and which does not have access to any alias table. As you point out, these results do lag behind the state of the art but we hope that our new experiments show what is responsible for the gap between the pure model's performance, and the state of the art.
>
> In response to your criticism of our vague characterisation of the CoNLL task, we have rewritten the entity linking section to focus on the specifics of how specialised entity linking approaches differ from the pure RELIC model. Thank you for this suggestion, we believe that the section is now much more robust and meaningful.
>
> We are glad that you found our investigation of the masking rates to be interesting and we would like to highlight that, simple as it is, masking mentions is a novelty of this paper not shared by previous work that focused on entity linking. Thank you also for your observation that RELIC's broad utility is hindered by the different optimum mask rates for entity linking and typing tasks. We have updated the paper with a discussion of optimum mask rates. We consider the choosing of an optimum mask rate to be an ongoing research question, and we plan to experiment with variable mask rates that account for variation in entity frequency.
>
> Finally, in response to your question about section 5.4, we have clarified the contents of Table 4. There are two settings for both the TypeNet and Wikipedia category completion tasks. The first contains all entities in each of those domains and the second (Yamada Subset) only contains the entities covered by the embedding table provided by Yamada et.al.  (https://github.com/studio-ousia/ntee). All of the comparisons between RELIC and Yamada use exactly the same entity and category vocabulary and the row for Yamada et.al. is left blank in the "All Entities" column because their embedding table does not contain all of the entities in this set. Since we perform a comparison to Yamada et.al. by using their provided embeddings, on their entity vocabulary, and with no further task specific training for either approach, we believe that this is the cleanest comparison that we can make between the RELIC and Yamada entity embedding tables on an entity typing task.

---

### Author Response · Authors · 2019-11-15
**Response to all reviews**

We thank the reviewers for their thorough reviews and constructive feedback.

We have responded below to each of the individual reviews. In this comment we address three topics of concern that were raised by multiple reviewers: (1) the simplicity of the RELIC model; (2) comparison to SOTA entity linking results; (3) how to compare our TriviaQA setup to previous work.

— Simplicity of the RELIC model —

We agree that the RELIC model is a simple adaptation of existing methods and we believe that this paper's main contribution is not the model, but the extensive and novel empirical evaluations.

Our Wikipedia category completion and trivia question answering experiments go far beyond previous work in demonstrating the extent to which distributed representations of entities can capture world knowledge that is generally stored in text (see discussion of TriviaQA results below) or hand built ontologies. We believe that this demonstration is worthy of publication in ICLR, notwithstanding the novelty of the model architecture.


— Comparison to entity linking state of the art —

Multiple reviewers asked about the gap between the pure RELIC model and the entity linking state of the art, and reviewer 3 correctly pointed out that we should back up our discussion of this task with a more robust reference to previous work.

In the original submission, we attempted to avoid any entity-linking specific enhancements to the RELIC model. However, to provide a cleaner comparison to previous work we have now made the following modifications, which are standard for the entity linking task:

  1. Limiting the CoNLL-AIDA search space with the widely used PPRforNED alias table from Pershina et.al. 2015.
  2. Reducing the TAC-KBP search space from the 5m entities covered by RELIC, to the 818k entities in the TAC-KBP knowledge
      base.
  3. Enhancing the RELIC contexts to always include the first 64 tokens of the current document.

With these three modifications, RELIC now achieves 94.9% accuracy on the CoNLL-Aida test set, matching the previous state of the art (DeepType, Raiman & Raiman 2018), and 89.8% on TAC-KBP which is just behind DeepType (90.9%) and ahead of all other approaches.

DeepType uses the extensive Wikidata knowledge base to build entity representations for use in linking. The fact that RELIC can match DeepType's performance with embeddings learned purely from text is significant, and these results support our central argument that RELIC is able to capture much of the knowledge that has previously been manually encoded in knowledge bases.

We would like to emphasise that even with the enhancements above, RELIC contains significantly fewer task specific components than all of the comparison systems. While we limited the TAC-KBP candidate set to the 818k entities in that domain, all other systems make use of an alias table to reduce it much further. Also, we note that Yamada et.al. 2016 used a few well chosen discrete features to boost performance of their embedding based model by 10%.  We omit these in favour of testing the limits of RELIC's purely learned representations.

— Comparison to previous work on TriviaQA —

Multiple reviewers enjoyed our re-casting of the TriviaQA task but pointed out that RELIC's performance is far below the upper bound on the more standard reading comprehension task.

We would like to re-emphasise that RELIC is performing a much harder task than the reading comprehension models. RELIC must retrieve the single correct answer from 5m candidate entities, while the reading comprehension models only need to consider spans in a single document (web setting).

A more recent and relevant comparison that we failed to make, is to Lee et.al. 2019. Like RELIC, Lee et.al.'s model (ORQA) is not given a single evidence document. Instead ORQA retrieves multiple passages from Wikipedia and then feeds these into a BERT based reading comprehension model.

On the open-domain TriviaQA-unfiltered setting, ORQA achieves 45% exact match accuracy compared to RELIC's 36%. While RELIC trails ORQA, it should be noted that RELIC's inference time computational requirements are massively lower. RELIC performs fast nearest neighbour search over the set of pre-indexed entity embeddings, while ORQA must run a deep transformer stack over multiple long evidence passages. Given these vast differences in answer time computational requirements, we consider it significant that RELIC captures 80% of ORQA's performance on this task.

We have updated the paper with a comparison to ORQA. We have also updated our discussion to more explicitly differentiate the TriviaQA reading comprehension task from the open-domain question answering task that ORQA and RELIC are attempting to solve.

—————————

Latent Retrieval for Weakly Supervised Open Domain Question Answering
Kenton Lee, Ming-Wei Chang, Kristina Toutanova (2019)
https://www.aclweb.org/anthology/P19-1612/

---

### Decision · Program_Chairs · 2019-12-19

**Decision:**

Reject

**Comment:**

The paper describes an approach for learning context dependent entity representations that encodes fine-grained entity types. The paper includes some good empirical results and observations, but the proposed approach is very simple but lacks technical novelty needed to top ML conference; the clarify of the presentation can also be improved.